# Litter Size of Sheep (*Ovis aries*): Inbreeding Depression and Homozygous Regions

**DOI:** 10.3390/genes12010109

**Published:** 2021-01-18

**Authors:** Lin Tao, Xiaoyun He, Xiangyu Wang, Ran Di, Mingxing Chu

**Affiliations:** Key Laboratory of Animal Genetics, Breeding and Reproduction of Ministry of Agriculture and Rural Affairs, Institute of Animal Science, Chinese Academy of Agricultural Sciences, Beijing 100193, China; 82101182357@caas.cn (L.T.); hexiaoyun@caas.cn (X.H.); wangxiangyu@caas.cn (X.W.); diran@caas.cn (R.D.)

**Keywords:** ROH, lambing number, sheep, inbreeding, fecundity

## Abstract

Ovine litter size (LS) is an important trait showing variability within breeds. It remains largely unknown whether inbreeding depression on LS exists based on genomic homozygous regions, and whether the homozygous regions resulted from inbreeding are significantly associated with LS in sheep. We here reanalyze a set of single nucleotide polymorphism (SNP) chip of six breeds to characterize the patterns of runs of homozygosity (ROH), to evaluate inbreeding levels and inbreeding depressions on LS, and to identify candidate homozygous regions responsible for LS. Consequently, unique ROH patterns were observed among six sheep populations. Inbreeding depression on LS was only found in Hu sheep, where a significant reduction of 0.016, 0.02, and 0.02 per 1% elevated inbreeding F_ROH4–8_, F_ROH_
_>_
_8_ and the total inbreeding measure was observed, respectively. Nine significantly homozygous regions were found for LS in Hu sheep, where some promising genes for LS possibly via regulation of the development of oocytes (*NGF*, *AKT1*, and *SYCP1*), fertilization (*SPAG17*, *MORC1*, *TDRD9*, *ZFYVE21*, *ADGRB3*, and *CKB*), embryo implantation (*PPP1R13B*, *INF2*, and *VANGL1*) and development (*DPPA2*, *DPPA4*, *CDCA4*, *CSDE1*, and *ADSSL1*), and reproductive health (*NRG3*, *BAG5*, *CKB*, and *XRCC3*) were identified. These results from the present study would provide insights into the genetic management and complementary understandings of LS in sheep.

## 1. Introduction

Litter size (LS), defined as the number of lambs born per ewe lambing, is of zoological and economical importance given the roles in survival of species and supply of products in sheep. LS is various within sheep breeds. Importantly, individual LS also varies among different parities. For instance, LS of 1–6 was recorded for Finnsheep, 1–5 for Romanov, 1–4 for Wadi, and 1–3 for Hu, Icelandic, and Texel sheep [1]. One possible explanation for the variability is the presence of inbreeding [2].

Inbreeding means the mating of related individuals, which is unfavorable in livestock due to the accompanying decrease on productive performances [3]. Inbreeding always extends the runs of homozygosity (referred to as ROH, homozygous segments) at the DNA level. ROH has been proposed to be used to calculate genomic inbreeding coefficients [4]. This measure was prevalent to evaluate inbreeding due to the consensus that it was more accurate using genomics data than only incorporating pedigree information [5]. Controlling inbreeding and inbreeding depression is a general goal in the management of livestock. However, the fact was that most inbreeding and inbreeding depression were evaluated only using pedigrees in sheep [6,7]. Hence, it would be interesting and necessary to use genomic information to evaluate the inbreeding coefficient and its effects on LS in different sheep breeds.

When the inbreeding depression on LS exists, the question then is, which homozygous regions resulted from inbreeding are associated with LS? As is well-known, single nucleotide polymorphisms (SNPs) have been widely used to identify quantitative trait loci (QTLs) and candidate genes from a genome-wide landscape [8]. The ability of homozygous regions to expose deleterious variants makes them a potential agency used in association analyses [9], which would make the best of the massive SNP chip for previous genome-wide association study (GWAS) based on SNPs. Recently, a growing body of interest was poured into the application of ROH in association with phenotypes in livestock [10,11]. Although some major genes such as *BMPR1B*, *GDF9*, and *BMP15* have been identified by GWAS or QTL mapping in some sheep populations [12], the relationship between genomic homozygous regions and LS remains largely unknown. Thereupon, genetic findings on homozygous regions related to inbreeding may provide complementary understandings and shed light on the mechanisms underlying LS in sheep.

Under the hypothesis that some homozygous regions, resulted from inbreeding and contributing to inbreeding depression on LS, may be significantly associated with LS, we reanalyzed a set of SNP chip from a previous GWAS to characterize the patterns of ROH, to evaluate inbreeding levels and inbreeding depressions on LS, and to identify candidate homozygous segments responsible for LS in six sheep breeds. These breeds excluding Texel are potential genetic resources to improve fecundity of other populations. These results would provide insights into genetic management and complementary understandings of LS in sheep.

## 2. Materials and Methods

### 2.1. Data

The data set used in this study were from a previous study [1] aiming at identifying candidate SNPs/genes for LS. Briefly, an Ovine Infinium HD BeadChip consisting of 606,006 SNPs was used for genotyping in six sheep breeds, of which based on the availability of both phenotype and genotype, we returned 314 unrelated ewes, including 100 Wadi, 77 Hu, 23 Icelandic, 37 Finnsheep, 38 Romanov and 39 Texel sheep. As for average litter size (ALS), the variability expressed as coefficients of variation (CV) within breeds is high (CV >15%, Table 1). According to the ALS (~82% with at least two lambing records) rank within the breeds as [1], each animal from two tails with extreme values was labelled with case (high-yield ewes) or control (low-yield ewes). Specifically, the cases included 77 Wadi (ALS ≥ 2), 62 Hu (ALS ≥ 2), 8 Icelandic (ALS > 2), 28 Finnsheep (ALS > 2.65), 29 Romanov (ALS > 2.65) and 28 Texel sheep (ALS ≥ 1.6), respectively. The controls contained 23 Wadi (ALS = 1), 15 Hu (ALS = 1), 15 Icelandic (ALS ≤ 1.75), 9 Finnsheep (ALS ≤ 2), 9 Romanov (ALS ≤ 1.8) and 11 Texel sheep (ALS < 1.3), respectively. Following the criteria of the original article, we performed quality control for each breed except a *p*-cutoff of 0.000001 for Hardy-Weinberg equilibrium considering the unique breed history. To explore the genetic relationships within each breed, pairwise identical by descent (IBD) was calculated, and principal component analysis (PCA) was performed using a pruning SNP data generated with the commend “indep-pairwise 50 5 0.2” in Plink 1.90 [13]. The R package ggplot2 (https://ggplot2-book.org/index.html) was used for PCA visualization.

### 2.2. ROH Calling and Inbreeding

For each breed, ROH was detected using Plink v1.90 [13] with an observational genotype-counting method. ROH was defined as a segment of (i) at least 1 Mb length, (ii) at least 100 consecutive homozygous SNPs, (iii) at least 1 SNP per 50 kb, (iv) at most 1 Mb of consecutive homozygous SNPs, and (v) at most one heterozygote and five missing calls.

In order to characterize the pattern of ROH within breeds, we calculated some descriptive statistics at individual level, including the number of ROH, and average and total length. We also calculated the genomic inbreeding coefficient F_ROH_, which equaled to the proportion of total ROH length on autosome 2655.71 Mb used in the present study. To further describe the effects of inbreeding on prolificacy, we performed simple linear regressions using F_ROH_ as an explanatory variable and ALS as our dependent variable with the linear model *ALS* = *β*_0_ + *β*_1_*F_ROH_* + *ε* where *β*_0_ was the intercept, *β*_1_ was the slope, and ε was the error. F_ROH_ was classified into three categories: (i) F_ROH1–4_ based on ROH of 1–4 Mb, (ii) F_ROH4–8_ based on ROH of 4–8 Mb, (iii) F_ROH > 8_ based on ROH with length of >8 Mb. Given the limited sample size, the descriptive statistics and inbreeding depression were performed for all individuals, not separately for case and control groups within each breed.

### 2.3. Genome-Wide ROH Hotspots Association Analysis

To explore the association between consensus homozygous segments and LS, we focused on the overlapping and potentially matching segments covering at least 10 SNPs and shared by at least 2 animals within each breed (here referred to as ROH hotspots). For simplicity, Fisher exact test was employed respectively to perform genome-wide case-control analysis (the presence and absence of ROH hotspots versus LS viz. high-yield or low-yield), followed by a Bonferroni correction (*p*-value = 0.05/the number of ROH hotspots tested) to control the false positive rate. Candidate ROH hotspots were mapped to the reference genome Oar_v.4.0 using ANNOVAR [14]. Metascape [15] was used to perform functional analyses for genes of interest, accepting a significance with a threshold of 0.05. Fisher exact test was also implemented to investigate the relationship between ALS and the variations within candidate ROH hotspots corrected by Bonferroni method using Plink v1.90 [13].

## 3. Results

### 3.1. Quality Control and Population Structure

A total of 308 individuals passed the quality control, including 100 Wadi (77 cases *vs*. 23 controls), 71 Hu (58 cases *vs*. 13 controls), 23 Icelandic (8 cases *vs*. 15 controls), 37 Finnsheep (28 cases *vs*. 9 controls), 38 Romanov (29 cases *vs*. 9 controls), and 39 Texel sheep (28 cases *vs*. 11 controls). Significant differences of LS within each breed were shown in Table 1. The number of markers that remained for ROH detection was over 440,000 for each breed (Table 1). The results of PCA showed the presence of slight population stratification (Appendix A). We repeated the following analyses without the outlying animals and got very similar results. Thus, we did not rule them out considering the small simple size and low average pairwise IBD (Wadi, 0.004; Hu, 0.002; Icelandic, 0.028; Finnsheep, 0.018; Romanov, 0.069; and Texel, 0.035) (Appendix A).

### 3.2. Pattern of ROH and Inbreeding Level

ROH features were characterized by number, and total and average length (Table 2). The number of ROH of Hu and Wadi sheep is much lower than that of other breeds (Icelandic, Finnsheep, Romanov and Texel). For the total length of ROH, which is in line with individual inbreeding level, Icelandic has the maximum (306.98 Mb), followed by Texel (258.17 Mb), whereas Wadi possesses the minimum (84.97 Mb). However, for average ROH length Hu features the longest (4.89 Mb) and Romanov the shortest (2.91 Mb). Collectively, unique ROH patterns were observed among these six breeds.

We also estimated the inbreeding coefficients for each breed (Appendix A and Table 3, Figure 1a). The extreme values (max 59.51% and min 0.10%) were in Hu population. Out of these ewes with F_ROH_ > 20% (22 animals), a large portion (16 animals, 73%) were Hu, followed by Wadi (four animals, 18%), Icelandic (one animal, 4.5%), and Finnsheep (one animal, 4.5%) (Figure 1). Among breeds, low inbreeding levels and a similar trend in line with total ROH length were observed. The regression analyses show that most effects are not significantly different from zero in the breeds excluding Hu sheep, and even the effect of F_ROH1–4_ is favorable in Romanov sheep (Table 3). However, a reduction of 0.16, 0.02, and 0.02 for ALS occurs as a result of per 1% elevated F_ROH4–8_, F_ROH > 8_ and the total inbreeding measure in Hu sheep, respectively (Table 3). These data imply the presence of inbreeding and inbreeding depression on ALS in Hu sheep.

### 3.3. Genome-Wide ROH Hotspots Association Analysis

Genome-wide association analyses based on ROH hotspots were performed for each breed to identify homozygous regions linked to LS (Figure 2). Unfortunately, no outliers were statistically observed in five populations including Wadi, Icelandic, Finnsheep, Romanov, and Texel sheep. However, nine ROH hotspots were identified for Hu sheep according to a threshold of 6.17 × 10^−5^ (Table 4). Accordingly, a total of 56 genes were found partly or entirely on these regions (Appendix A). These genes were functionally grouped into four pathways, including the PID PI3K PLC TRK pathway, apoptotic signaling pathway, response to calcium ion, and cellular process involved in reproduction in multicellular organism (Figure 3 and Appendix A). The relevance of candidate genes on reproduction was determined by a literature search. Consequently, straightforward connections between LS and two candidate genes (*PPP1R13B* and *NGF*) have been reported [16,17]. Additionally, the potential reproductive roles of candidate genes in development of oocytes and sperm, zygotic transcriptional program, embryo implantation and development, and reproductive diseases were uncovered (Figure 4).

To investigate promising variations at above regions, Fisher exact tests were implemented to explore the relationship between SNPs and ALS. Manhattan plots for each ROH hotspots of interest were shown in Appendix A. Only one intergenic marker on S325 was statistically significant, which was in the proximity of *DPPA4* and *DPPA2* (Appendix A).

## 4. Discussion

### 4.1. ROH and Inbreeding

In this study, we characterized the patterns of homozygosity in six sheep breeds, including two Chinese breeds and four non-Chinese breeds. Unique ROH patterns were observed, which would be attributed to their population history and selective breeding. A low genomic inbreeding coefficient (0.065) of Finnsheep was evaluated in the present study, in line with an evaluation from a previous study (0.060) [18]. F_ROH_ of 0.093 was estimated for Hu sheep breed, which was larger than the previous value (0.048) using the same method [18]. One possible explanation for this discrepancy is different sample size and density of SNP chip. Interestingly, only the inbreeding level of Wadi sheep (0.032, equals to the average of 41 Chinese breeds) is lower compared with the global average (0.046) derived from 1910 animals in 97 populations [18]. For Wadi sheep, the low inbreeding means an effective genetic management. However, for other five breeds measures should be taken to control inbreeding.

Unfavorable effects of inbreeding on LS were observed in Hu sheep, which coincides with the inbreeding depressions of Danish and Czech populations based on pedigree [7,19]. It is important to note that Hu sheep take a large portion (16/22) of these ewes with a high inbreeding level (F_ROH_ > 0.2), of which the maxima is about 0.6. This coincides with the history of mother-son mating as long as at least 900 years in Hu sheep [20]. Specifically, short ROH (1–4 Mb) can represent ancient inbreeding, and the unfavorable effect of F_ROH1–4_ on ALS was supported by the history of mother-son mating. On the other hand, the recent inbreeding resulted in long ROH (>8 Mb) and the significant decrease of ALS suggested recently ineffective genetic management in Hu sheep. The ancient and recent inbreeding highlighted the urgence of measures to maintain high fecundity and limit inbreeding in Hu sheep. However, these results are from a small herd and more animals are needed in future studies.

### 4.2. Candidate Genes Likely Functioning in Prolificacy

Significant homozygous regions were found for LS in Hu sheep rather than the others in the present study, which supported our hypothesis that some homozygous regions contributing to inbreeding depression on LS, may be significantly associated with LS. From the reproductive standpoint of ewes, LS is the outcome of a series of physiological processes, including the development of oocytes, fertilization, embryo implantation and development [21,22]. Hence, more or less effects of genes associated with these processes were expected on the ending of ovine delivery, supported (at least partly) by these candidate genes identified in Hu sheep. Although previous prolificacy genes did not emerge from the present study, some function-associated genes were identified (Figure 4).

Straightforward evidence was well-documented that *PPP1R13B* and *NGF* influenced LS. Loss of *PPP1R13B*, also known as *p53*, female mice decreased significantly embryonic implantation, pregnancy rate and LS via regulating leukaemia inhibitory factor, which was a cytokine critical for implantation [16,23]. In pigs, 17β-estradiol, one important reproductive hormone, was found to promote transcription of *p53* [24]. Interestingly, the polymorphism of *NGF*, playing a role in augmenting folliculogenesis, has been reported to be associated with LS in both sheep and goats [17,25,26,27].

AKT1, one isoform of AKT (a serine/threonine kinase), has been extensively studied in reproductive processes through phosphoinositol 1,3 kinase/protein kinase B (PI3K/AKT) signaling, including follicular development and embryo implantation [28,29,30]. The roles of PI3K/AKT signaling in follicular development consist of (i) the initiation of primordial follicle growth via suppresses FOXO3 (a transcriptional factor keeping primordial follicle dormancy) actions, (ii) the regulation in granulosa cell differentiation of antral follicles via FSH (follicle-stimulating hormone), and (iii) the participation in oocyte maturation of preovulatory follicles [31,32,33]. In addition, SYCP1 is required for the formation of crossovers during meiotic prophase [34], indicating its role in the development of oocytes.

Embryo implantation and development is another important reproductive event influencing LS. Zygotic genome activation launches the expression of parental genomes, during which any wrongdoing may terminate embryo development [35]. DPPA2 and DPPA4 were involved in zygotic genome activation by regulating *Dux* and LINE-1 retrotransposons [36,37]. Protein inhibitor of activated STAT 4 (PIAS4) and histone methyltransferase SETDB1 could negatively modulate DPPA2 protein activity [35,38]. Late embryonic/perinatal death was also observed in Dppa4-deficient mice [39]. Furthermore, Cdca4 was identified as a target gene of miR-154 differently expressed between 2-cell and 4-cell mouse embryos [40]. The placenta is an organ linking baby and mom during pregnancy, on which proliferation, differentiation, and invasion of trophoblast cells are vital to healthy pregnancy. Both maternal and fetal phenotypes of placental insufficiency were observed in the *Inf2*-lacking mice [41], highlighting the essential roles of *Inf2* during pregnancy. Vangl1, a principal component of Wnt/PCP signaling pathway, is required for embryonic stem cells (ESC) differentiation, early embryo development and embryo implantation [42,43]. Highly expressed in human ESC, CSDE1 was known as a central post-transcriptional regulator of ESC identity and neurogenesis [44]. *ADSSL1* was identified as a candidate gene for fetal akinesia [45]. Collectively, these candidate genes play crucial roles in embryo implantation and development.

Health is an important reproductive content. Of interest, some candidates related to sperm and infertility were identified, containing *MORC1* [46], *TDRD9* [47], *ZFYVE21* [48], *ADGRB3* [49], *SPAG17* [50,51], *CKB* [52], and *WDR3* [53], which may have effects on sperm motility and fertilization. Additionally, we identified some candidate genes linked to reproductive diseases, such as *NRG3* [54] and *XRCC3* [55] for ovarian cancer, and *BAG5* [56], *CKB* [57] and *XRCC3* [55] for endometriosis. Taken together, these data indicate the importance of reproductive health in lambing.

## 5. Conclusions

The present study grounded on an SNP chip explored genome-wide homozygosity in six sheep flocks and pinpointed the presence of inbreeding depression for LS in Hu sheep. Additionally, based on genome-wide ROH hotspots associated analysis, this study identified some promising genes for LS possibly via regulation of the development of oocytes (*NGF*, *AKT1*, and *SYCP1*), fertilization (*SPAG17*, *MORC1*, *TDRD9*, *ZFYVE21*, *ADGRB3*, and *CKB*), embryo implantation (*PPP1R13B*, *INF2*, and *VANGL1*) and development (*DPPA2*, *DPPA4*, *CDCA4*, *CSDE1*, and *ADSSL1*), and reproductive health (*NRG3*, *BAG5*, *CKB*, and *XRCC3*). These results would provide insights into the genetic management and complementary understandings of LS in sheep.

## Figures and Tables

**Figure 1 genes-12-00109-f001:**
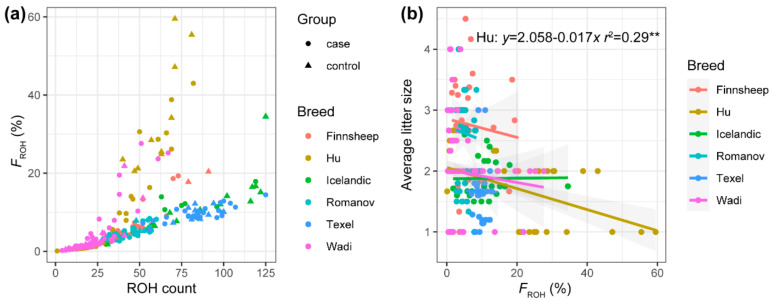
Inbreeding levels of six sheep breeds. Scatter plots of F_ROH_ against the number of runs of homozygosity (ROH) (**a**), and average litter size against F_ROH_ (**b**). ** denotes *p*-value < 0.01. The regression for Hu sheep is shown because it is the only breed with significant coefficients.

**Figure 2 genes-12-00109-f002:**
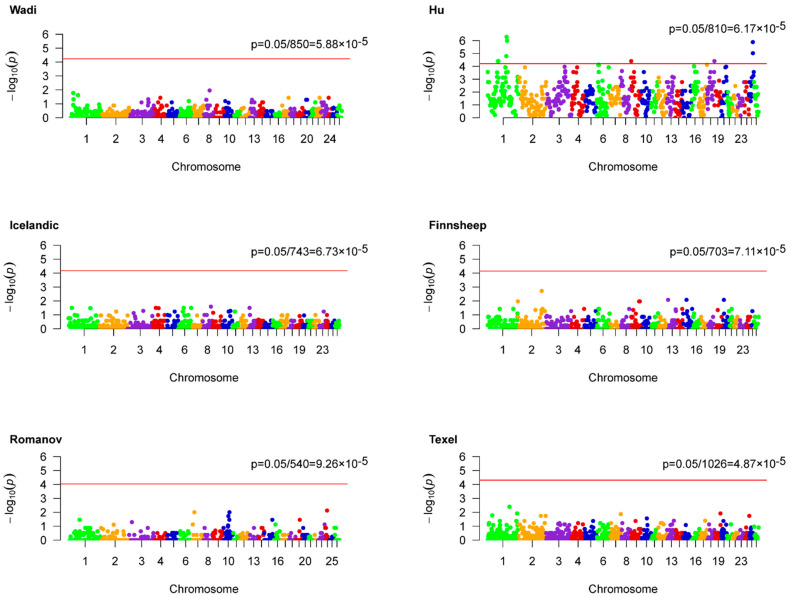
Genome-wide ROH hotspots associated studies for litter size in six sheep breeds. The red lines denote the thresholds used to extra outliers.

**Figure 3 genes-12-00109-f003:**
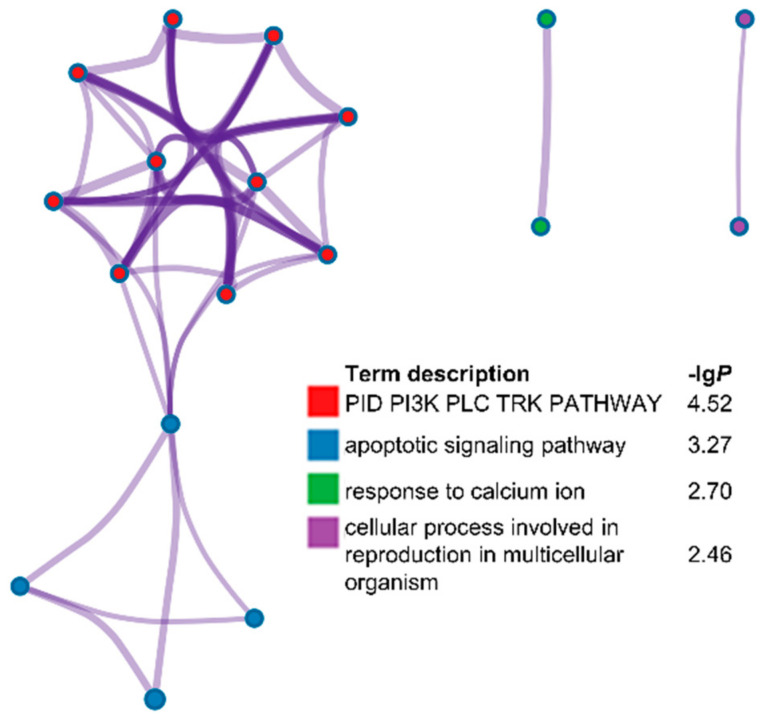
Function annotation of candidate genes. A total of 26 child-terms were returned which were summarized into four parent-terms (Appendix A).

**Figure 4 genes-12-00109-f004:**
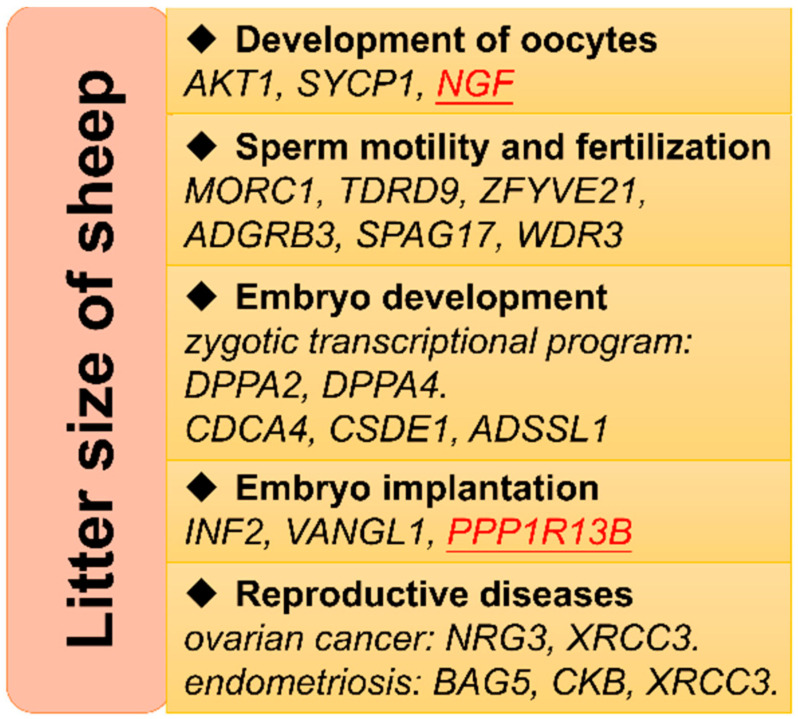
Some possible biological processes, by which candidate genes influence litter size in sheep. The underlined genes in red have been reported to be associated with litter size.

**Table 1 genes-12-00109-t001:** Genotypes and phenotypes of six sheep breeds.

Breed	SNP for ROH Calling	Average Litter Size (Mean ± SD) ^†^			CV (%)
		Case	Control	Total	
Wadi	512,178	2.28 ± 0.50 ^A^ (n = 77)	1.87 ± 0.10 ^B^ (n = 28)	1.99 ± 0.65 (n = 100)	32.81
Hu	503,422	2.10 ± 0.24 ^A^ (n = 58)	1.00 ± 0.00 ^B^ (n = 13)	1.90 ± 0.47 (n = 71)	24.87
Icelandic	443,125	2.29 ± 0.19 ^A^ (n = 8)	1.66 ± 0.10 ^B^ (n = 15)	1.88 ± 0.33 (n = 23)	17.83
Finnsheep	492,165	3.11 ± 0.45 ^A^ (n = 28)	1.66 ± 0.33 ^B^ (n = 9)	2.76 ± 0.76 (n = 37)	27.55
Romanov	465,794	2.97 ± 0.29 ^A^ (n = 29)	1.53 ± 0.25 ^B^ (n = 9)	2.63 ± 0.68 (n = 38)	25.90
Texel	475,999	1.87 ± 0.35 ^A^ (n = 28)	1.15 ± 0.10 ^B^ (n = 11)	1.66 ± 0.45 (n = 39)	26.86

^†^ For each breed, different letters denote highly significant difference (*p* < 0.01) using the Wilcoxon rank sum test with continuity correction. The sample sizes were in brackets. Ranked by the average litter size, each animal from two tails with extreme values was labelled with case (high-yield ewes) or control (low-yield ewes).

**Table 2 genes-12-00109-t002:** Runs of homozygosity (ROH) feature of six sheep breeds.

ROH Feature	Breed ^†^					
	Hu	Wadi	Icelandic	Finnsheep	Romanov	Texel
Number						
Average	29.86 ^C^	21.26 ^C^	81.96 ^A^	45.38 ^B^	43.71 ^B^	85.03 ^A^
SD	20.90	11.05	29.07	15.76	9.01	14.42
Maximum	82	67	125	91	60	125
Total length (Mb)						
Average	247.9 ^C^	84.97 ^C^	306.98 ^A^	173.51 ^B^	128.15 ^B^	258.17 ^A^
SD	388.94	140.37	175.46	131.91	41.40	47.75
Maximum	1580.38	733.08	916.21	541.73	223.37	384.18
Average length (Mb)						
Average	4.89 ^B^	3.01 ^B^	3.58 ^A^	3.48 ^A^	2.91 ^A^	3.06 ^A^
SD	5.42	2.64	1.17	1.36	0.59	0.41
Maximum	22.26	14.37	7.33	7.06	4.32	3.97

^†^ For each ROH feature, breeds with different letters denote highly significant difference (*p* < 0.01, Kruskal-Wallis test) with multiple comparisons using the Wilcoxon rank sum test.

**Table 3 genes-12-00109-t003:** Estimates of inbreeding depression for average litter size, expressed as the change in expected phenotype per 1% increase inbreeding.

F_ROH_	Estimate ± SE					
	Wadi	Hu	Icelandic	Finnsheep	Romanov	Texel
F_ROH1–4_	0.12 ± 0.16	−0.22 ± 0.13	−0.03 ± 0.04	−0.20 ± 0.20	0.35 ± 0.23	−0.02 ± 0.08
F_ROH4–8_	−0.15 ± 0.14	−0.16 ** ± 0.05	−0.01 ± 0.05	−0.08 ± 0.12	−0.06 ± 0.19	−0.11 ± 0.10
F_ROH > 8_	−0.01 ± 0.01	−0.02 ** ± 0.00	0.01 ± 0.01	−0.01 ± 0.03	−0.11 ± 0.11	0.04 ± 0.06
All	−0.01 ± 0.01	−0.02 ** ± 0.00	0.00 ± 0.01	−0.02 ± 0.03	−0.02 ± 0.07	0.00 ± 0.04

** denotes *p* < 0.01. The values without ** were not significant (*p* > 0.05).

**Table 4 genes-12-00109-t004:** Nine ROH hotspots significantly linked to litter size in Hu sheep.

ROH Hotspot	Region ^†^	Length (kb)	N_SNP	N_Gene	*p*-Value
S32	1:167605062-167736007	130.946	24	0	5.14 × 10^−7^
S325	1:171893678-173273898	1380.221	269	5	1.01 × 10^−6^
S736	25:34831274-34920313	89.04	20	1	1.29 × 10^−6^
S599	25:35553072-36870587	1317.516	275	2	9.48 × 10^−6^
S125	1:164718392-165322668	604.277	107	1	1.60 × 10^−5^
S414	18:66738545-68488609	1750.065	299	30	3.91 × 10^−5^
S378	9:5081783-5791789	710.007	161	1	3.91 × 10^−5^
S312	1:94177534-94744969	567.436	129	3	3.91 × 10^−5^
S311	1:90921006-92192152	1271.147	227	13	3.91 × 10^−5^

^†^ The region covering the bases form *m*th to *n*th at chromosome *l* was denoted as *l:m-n*. N_SNP and N_Gene are the number of single nucleotide polymorphisms (SNP) and gene located at ROH hotspots, respectively.

## Data Availability

Genotype and phenotype datasets were available at the following link: https://www.animalgenome.org/repository/pub/CAAS2018.0302/.

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
