# Peer review of "Litter Size of Sheep (Ovis aries): Inbreeding Depression and Homozygous Regions"

_genes, 2021, doi:10.3390/genes12010109_

Round 1

Reviewer 1 Report

Dear Authors,

the manuscript “Litter size of sheep (Ovis aries): inbreeding depression and homozygous loci” deals with the analysis of the relationship between inbreeding levels and litter size in six different sheep breeds.

The paper was well written, however, I have a main doubt about the model you used to test whether inbreeding depression exits in the six sheep breeds. Moreover, I found also some other points that need to be fixed.

The main concern is about the model you used to estimate inbreeding depression. I would like to know why you did not apply a linear model with more effects: I don’t feel confident that LS changes only according to the ROH inbreeding coefficient. I suppose that the ewes came from different herds, were at different parities and were sired by different rams.

I would have tried a more complex model, with at least parity (or age) as explanatory variable; I would have evaluated also the use of sire as random effect, since I think that it would have a not ignorable effect on LS.

This doubt is somehow confirmed also by results reported in Table 3, where several values have SE larger than estimates (so no different from 0) and some are positive (so no inbreeding depression, but positive effect of inbreeding). The value estimated for Romanov using FROH 1-4 is 0.35(0.23) meaning a positive effect of inbreeding on litter size; right?. Based on these results, I don’t fully support your thesis about inbreeding depression in the investigated sheep breeds. You need to clearly state that inbreeding depression was found only in Hu sheep breed.

Line-by-line comments:

Line 9: I suggest changing “is an important trait and exists variability within breeds” with “is an important trait showing variability within breeds”

Lines 30-33: these lines seem more materials and methods than introduction; you are reporting here numbers of the dataset used in this study, so please move these lines to M&M. You can replace them with a connection between litter size and the inbreeding you present from line 34.

Line 46: how we can say that “homozygous loci” resulted from inbreeding?

Lines 55: “the relationship between of genomic …” please remove of

Line 66: Were the animals within breed related each-other? Did the animals come from more than one herd? Please add some details about the samples.

Lines 72-73: how did you define the tails? I supposed you used quartiles but then at lines 115-116 the number of animals belonging to case and controls are very different (e.g., 58 cases and 13 controls for Hu).

Line 88: why did you use a so high number of missing allowed in the ROH?

Line 125, Table 1: please remove bold from “Wadi” and from “512,178”; please also remove the line below “Wadi”.

Lines 144-145: are these percentages of total animals with FROH>20%? Or percentage of animals belonging to each breed? I guess the first one because for example for Hu 16 animals / 71 animals is just 22% not 73%. If the percentages refer to animals with FROH > 20% please include the total number of these animals in “out of these with FROH > 20% (XX)” at line 144.

Lines 147-148: I don’t fully support this sentence about “… negative effects of inbreeding on ALS for all breeds”. In table 3 we have positive values (e.g., Romanov FROH 1-4; meaning that some kind of inbreeding is having a positive effect on LS) and null ones (e.g., Iceland with All ROH). Moreover, lots of values have larger SE than the estimates, so we cannot consider them different than zero. Please mention this when speaking about Table 3.

Line 153, Table 3: Please remove “-” before the 0.00 in “Texel All”. Were the values without * not significant? Or just significant with P-value < 0.05?

Line 156, Figure 1: Why did you report just the regression for Hu? Because it was the only breed with significant coefficients?

Line 163: Please consider changing “loci” with “regions”.

Line 175, Table 4: Is ROH hotspot just a progressive number or it has a significance?

Lines 194-195: “Hu sheep was estimated with FROH …” seems not quite English and could be written as “FROH of 0.0.93 was estimated for Hu sheep breed….”

Line 195: “which was larger more than the calculation…” please rephrase it; for example, “which was larger than previous value (0.048) estimated using the same method” or something like this.

Line 198: where did you find the information that 0.032 is the average value of 41 Chinese breeds? Please add a reference.

Line 204: “It is important to note that Hu sheep take a large portion (16/22) of these”, these what?

Line 214: The whole paragraph 4.2 needs to be carefully revised; several sentences are hard to follow.

Lines 268-269: As aforementioned, I don’t fully support your thesis that inbreeding depression is affecting LS in sheep.

Table S1: The numbers in Table S1 seem weird to me. The values in FROH 4-8 are lower than these in FROH>8 because you had few ROHs in the class 4-8? Just a comment, probably it was better to use FROH>1, FROH>4 and FROH>8? In this way you can have a trend, because the previous class contains also the following one.

Author Response

Responds to Reviewer 1

Dear Authors,

the manuscript “Litter size of sheep (Ovis aries): inbreeding depression and homozygous loci” deals with the analysis of the relationship between inbreeding levels and litter size in six different sheep breeds.

The paper was well written, however, I have a main doubt about the model you used to test whether inbreeding depression exits in the six sheep breeds. Moreover, I found also some other points that need to be fixed.

Reply: We sincerely thank you for your time spent on reviewing our manuscript. We appreciate your useful comments, which have largely helped us to improve our manuscript. Please see our revisions according to your comments in our revised manuscript, and our responses to your comments as follows.

Tips: It seems that the manuscript you reviewed is different from the one we can access. The difference is at least one line after the line 66.

The main concern is about the model you used to estimate inbreeding depression. I would like to know why you did not apply a linear model with more effects: I don’t feel confident that LS changes only according to the ROH inbreeding coefficient. I suppose that the ewes came from different herds, were at different parities and were sired by different rams.

I would have tried a more complex model, with at least parity (or age) as explanatory variable; I would have evaluated also the use of sire as random effect, since I think that it would have a not ignorable effect on LS.

Reply: Thank you for your good comments.

We totally agree with your idea that more potential factors should be considered in the model to estimate inbreeding depression. However, unfortunately, the information (including the herds and rams) for each breed is publicly unavailable. We excluded the effect of parity from the model because (1) its effect is not significant for the ewes involved in this study within each breed, and (2) we focused on the average litter size due to different parities of individuals within breeds. Therefore, a simple linear regression was used in this study. We think our results of inbreeding depression are reasonable, which is observed in Hu sheep, not the remaining breeds. This is supported by the association studies using ROH hotspots, in which significant ROH hotspots are only identified in Hu sheep. In other word, the presence of inbreeding depression ensures identification of significant ROH segments in Hu sheep.

Thanks again for your insightful suggestions.

This doubt is somehow confirmed also by results reported in Table 3, where several values have SE larger than estimates (so no different from 0) and some are positive (so no inbreeding depression, but positive effect of inbreeding). The value estimated for Romanov using FROH 1-4 is 0.35(0.23) meaning a positive effect of inbreeding on litter size; right?. Based on these results, I don’t fully support your thesis about inbreeding depression in the investigated sheep breeds. You need to clearly state that inbreeding depression was found only in Hu sheep breed.

Reply: Thank you for pointing it out.

We acknowledge that several values have SE larger than estimates in Table 3, which is in line with the statistical non-significance, indicating the absence of inbreeding depression in the breeds excluding Hu sheep. However, it is important to note that the phenomenon that SE is larger than estimate in the measurement of inbreeding depression is a frequent occurrence. For example, although complex fixed and random effects were considered in models in a recent study, the abovementioned phenomenon was still observed (Table 3 and Figures 2-5) (Makanjuola et al. 2020). Therefore, we argue that our results in Table 3 are reasonable and acceptable.

As for the thesis, we totally agree with you. In the Abstract, Results, Discussion and Conclusions of our old manuscript, we just stated that inbreeding depression was observed in Hu sheep, not others. This is in concordance with what you proposed and suggested. Now we describe it more clearly based on your suggestions. Please see the manuscript for details.

Line-by-line comments:

Line 9: I suggest changing “is an important trait and exists variability within breeds” with “is an important trait showing variability within breeds”

Reply: Thanks! We changed it as your suggestion.

Lines 30-33: these lines seem more materials and methods than introduction; you are reporting here numbers of the dataset used in this study, so please move these lines to M&M. You can replace them with a connection between litter size and the inbreeding you present from line 34.

Reply: Thank you! We move this sentence to M&M according to your advice.

Line 46: how we can say that “homozygous loci” resulted from inbreeding?

Reply: We are sorry for this confusion. What we want to express is inbreeding can result in homozygous regions. That is, homozygous regions are resulted from inbreeding.

Lines 55: “the relationship between of genomic …” please remove of

Reply: Thank you! We changed it as your suggestion.

Line 66: Were the animals within breed related each-other? Did the animals come from more than one herd? Please add some details about the samples.

Reply: Thanks. Animals included were as unrelated as possible based on analysis of pedigree records and farmers’ knowledge (Xu et al. 2018). It is also confirmed by the very low pairwise identical by descent (IBD) (Figure S1). The authors did not mention whether the animals came from more than one herd (Xu et al. 2018). We try to make a balance between clear description on samples and avoiding repetition, considering the availability of the details in a previous study (Xu et al. 2018).

Lines 72-73: how did you define the tails? I supposed you used quartiles but then at lines 115-116 the number of animals belonging to case and controls are very different (e.g., 58 cases and 13 controls for Hu).

Reply: Thank you. Specifically, the cases include 77 Wadi (average litter size, ALS ≥ 2), 62 Hu (ALS ≥ 2), 8 Icelandic (ALS > 2), 28 Finnsheep (ALS > 2.65), 29 Romanov (ALS > 2.65) and 28 Texel sheep (ALS ≥ 1.6), respectively. And the controls contain 238 Wadi (ALS = 1), 15 Hu (ALS = 1), 15 Icelandic (ALS ≤ 1.75), 9 Finnsheep (ALS ≤ 2), 9 Romanov (ALS ≤ 1.8) and 11 Texel sheep (ALS < 1.3), respectively.

Line 88: why did you use a so high number of missing allowed in the ROH?

Reply: Because (1) the marker density of the SNP chip is high, and (2) it is a default parameter.

Line 125, Table 1: please remove bold from “Wadi” and from “512,178”; please also remove the line below “Wadi”.

Reply: Thanks for your carefulness! We changed it according to your suggestion.

Lines 144-145: are these percentages of total animals with FROH>20%? Or percentage of animals belonging to each breed? I guess the first one because for example for Hu 16 animals / 71 animals is just 22% not 73%. If the percentages refer to animals with FROH > 20% please include the total number of these animals in “out of these with FROH > 20% (XX)” at line 144.

Reply: Thank you for your suggestion. The former is the case, so we add the total number of animals.

out of these ewes with FROH > 20% (22 animals)

Lines 147-148: I don’t fully support this sentence about “… negative effects of inbreeding on ALS for all breeds”. In table 3 we have positive values (e.g., Romanov FROH 1-4; meaning that some kind of inbreeding is having a positive effect on LS) and null ones (e.g., Iceland with All ROH). Moreover, lots of values have larger SE than the estimates, so we cannot consider them different than zero. Please mention this when speaking about Table 3.

Reply: Thank you very much!

We add the points you suggested to the revised manuscript. Please see the revised version.

Line 153, Table 3: Please remove “-” before the 0.00 in “Texel All”. Were the values without * not significant? Or just significant with P-value < 0.05?

Reply: Thanks. The values without * were not significant (P>0.05). We now describe it more clearly.

Line 156, Figure 1: Why did you report just the regression for Hu? Because it was the only breed with significant coefficients?

Reply: Because it is significant only for Hu sheep (P<0.01).

Line 163: Please consider changing “loci” with “regions”.

Reply: Thank you very much. We agree with you and change it.

Line 175, Table 4: Is ROH hotspot just a progressive number or it has a significance?

Reply: The nine ROH hotspots were significantly associated with litter size in Hu sheep by Fisher exact test.

Lines 194-195: “Hu sheep was estimated with FROH …” seems not quite English and could be written as “FROH of 0.0.93 was estimated for Hu sheep breed….”

Reply: Thanks. We rewrite it as your suggestion.

Line 195: “which was larger more than the calculation…” please rephrase it; for example, “which was larger than previous value (0.048) estimated using the same method” or something like this.

Reply: Thanks! We rephase it.

Line 198: where did you find the information that 0.032 is the average value of 41 Chinese breeds? Please add a reference.

Reply: The value (0.032) of 41 Chinese breeds is from Table S1 of the work (Zhang et al. 2018).

Line 204: “It is important to note that Hu sheep take a large portion (16/22) of these”, these what?

Reply: It is important to note that Hu sheep take a large portion (16/22) of these ewes with a high inbreeding level (FROH > 0.2), of which the maxima is about 0.6.

Line 214: The whole paragraph 4.2 needs to be carefully revised; several sentences are hard to follow.

Reply: Thank you. We checked and revised it as careful as possible. Please kindly point them out if some sentences are still hard to follow.

Lines 268-269: As aforementioned, I don’t fully support your thesis that inbreeding depression is affecting LS in sheep.

Reply: Thanks. We agree with you. Our thesis that the inbreeding depression on litter size is limited to the Hu sheep population is consistent with yours.

Table S1: The numbers in Table S1 seem weird to me. The values in FROH 4-8 are lower than these in FROH>8 because you had few ROHs in the class 4-8? Just a comment, probably it was better to use FROH>1, FROH>4 and FROH>8? In this way you can have a trend, because the previous class contains also the following one.

Reply: Thank you for your suggestions.

In fact, two factors would determine the estimate of inbreeding coefficients, including (1) the number of ROH, and (2) the length of ROH.

The method you proposed to estimate inbreeding is very good because it can provide a trend of inbreeding level. However, the method we used aims to estimate inbreeding from different ages through the length of ROH, because inbreeding depression could be distinctly associated with more recent or more ancient inbreeding (Makanjuola et al. 2020). Inferring the age of inbreeding from the length of ROH segment is an expectation that follows an exponential distribution with a mean of 100/2g centiMorgans (cM), where g is the number of generations to a common ancestor (Druet & Gautier 2017). In principle, the two methods are equivalent.

Thank you again.

References

Druet T. & Gautier M. (2017) A model‐based approach to characterize individual inbreeding at both global and local genomic scales. Molecular Ecology 26, 5820-41.

Makanjuola B.O., Maltecca C., Miglior F., Schenkel F.S. & Baes C.F. (2020) Effect of recent and ancient inbreeding on production and fertility traits in Canadian Holsteins. BMC Genomics 21, 605.

Xu S.S., Gao L., Xie X.L., Ren Y.L., Shen Z.Q., Wang F., Shen M., Eyϸórsdóttir E., Hallsson J.H., Kiseleva T., Kantanen J. & Li M.H. (2018) Genome-wide association analyses highlight the potential for different genetic mechanisms for litter size among sheep breeds. Frontiers in Genetics 9, 118.

Zhang M., Peng W.F., Hu X.J., Zhao Y.X., Lv F.H. & Yang J. (2018) Global genomic diversity and conservation priorities for domestic animals are associated with the economies of their regions of origin. Scientific Reports 8, 11677.

Reviewer 2 Report

The authors presented a paper on litter size of sheep and effect of inbreeding depression and homozygous loci. The paper is interesting and well conducted.

There are some points of concern that need authors solve out.

More in details:

title: I suggest to improve the title as follows: Litter size of sheep (Ovis aries): effect of inbreeding depression and homozygous loci.

Abstract: if possible try to reduce the use of acronyms in this section.

Keywords: it is better to avoid words already included in the title.

Introduction:

-the introduction section is useful and complete. However, it seems subdivided in three different topics (litter size, ROH and GWAS): if possible please try to make this section more fluently for the readers.

-finally it could be better to explain, at the end of this section, why the authors decided to study these 6 breeds, supplying also general information about the studied breeds (present census, productive purpose, etc).

Material and methods

It could be usueful for the readers to have some information about the kind of biological samples collected for the DNA extraction.

In the data quality control did the authors used parameters such as call rate or minor allele frequency (MAF)? Please specify.

Please specify the software used to carry out the gene ontology.

Line 76: please replace 28 Wadi with 23 Wadi.

Line 82: what software was used for the graphical PCA representation?

Results

Lines 115-118, please use italic style for “vs”.

Line 153: please replace “in breeding” with “inbreeding”.

Why the authors after the Bonferroni corrections are using the P-value and not the q-value (that I think it is used in the Manhattan plots): it could be better to use the same values both in GWAS and CWAS.

Discussion

Lines 217-220: this sentence needs a reference.

Author Response

Responds to Reviewer 2

The authors presented a paper on litter size of sheep and effect of inbreeding depression and homozygous loci. The paper is interesting and well conducted.

There are some points of concern that need authors solve out.

Reply: Thank you for your time spent on reviewing our manuscript. We sincerely appreciate your valuable comments which have definitely helped us to improve our manuscript. Please see our revised manuscript and our responses to your comments in the following.

Tips: It seems that the manuscript you reviewed is different from the one we can access. The difference is at least one line after the line 66.

More in details:

title: I suggest to improve the title as follows: Litter size of sheep (Ovis aries): effect of inbreeding depression and homozygous loci.

Reply: Thanks. The title you proposed highlights the effect of inbreeding depression. We politely think that inbreeding depression actually is an effect of inbreeding on a given trait (here litter size), so it may be redundant to use the word ‘effect’. After careful consideration, we would like to recommend the following one:

Litter size of sheep (Ovis aries): inbreeding depression and homozygous regions.

Abstract: if possible try to reduce the use of acronyms in this section.

Reply: Thank you. Considering the frequent occurrence of litter size (LS) and runs of homozygosity (ROH), we abbreviated them. For conciseness, abbreviations were used for genes.

Keywords: it is better to avoid words already included in the title.

Reply: Thank you very much. We replace litter size by lambing number.

Introduction:

-the introduction section is useful and complete. However, it seems subdivided in three different topics (litter size, ROH and GWAS): if possible please try to make this section more fluently for the readers.

Reply: Thank you. We make some revisions according to Reviewer 1’s suggestion. Please see the revised manuscript for details.

-finally it could be better to explain, at the end of this section, why the authors decided to study these 6 breeds, supplying also general information about the studied breeds (present census, productive purpose, etc).

Reply: Thanks. We used these six breeds because (1) the variability expressed as coefficients of variation (CV) within breeds is high for average litter size (CV >15%, Table 1); (2) the dataset of phenotype and genotype of them are available; and (3) these breeds excluding Texel are potential genetic resources to improve fecundity of other populations.

Material and methods

It could be useful for the readers to have some information about the kind of biological samples collected for the DNA extraction.

Reply: Thanks. It is important to note that this study is based on a dataset available in a previous study (Xu et al. 2018), where the information on biological samples (whole-blood for Icelandic sheep and ear marginal tissue for other five breeds) has been described. So, we did not mention it in the present study.

In the data quality control did the authors used parameters such as call rate or minor allele frequency (MAF)? Please specify.

Reply: Thanks. What you mentioned has been described in Lines 77-78.

Following the criteria of the original article (Xu et al. 2018), we performed quality control for each breed except a P-cutoff of 0.000001 for Hardy-Weinberg equilibrium considering the unique breed history.

Please specify the software used to carry out the gene ontology.

Reply: Metascape (Zhou et al. 2019) has been mentioned for functional analyses (including gene ontology) in the section 2.3 of our old manuscript.

Line 76: please replace 28 Wadi with 23 Wadi.

Reply: Thanks for your carefulness. We corrected it.

Line 82: what software was used for the graphical PCA representation?

Reply: The R package ggplot2 (https://ggplot2-book.org/index.html) was used for PCA visualization.

Results

Lines 115-118, please use italic style for “vs”.

Reply: Thanks. We correct it in the revised manuscript.

Line 153: please replace “in breeding” with “inbreeding”.

Reply: Thank you very much. We change it according to your suggestion.

Why the authors after the Bonferroni corrections are using the P-value and not the q-value (that I think it is used in the Manhattan plots): it could be better to use the same values both in GWAS and CWAS.

Reply: We are sorry for not following your comments. What is the CWAS?

For each sheep breed, p-value adjusted by a Bonferroni method was used to extract outliers in both Figure 2 and Table 4. If you mean why different cutoff of adjusted p-values are used for six breeds, that because the number of marker (ROH hotspots) used for association analyses is different.

Thank you anyway.

Discussion

Lines 217-220: this sentence needs a reference.

Reply: Thanks. We now add the references (Hanrahan & Quirke 1984; Haresign 1985).

References

Hanrahan J. & Quirke J. (1984) Contribution of variation in ovulation rate and embryo survival to within breed variation in litter size. Genetics of reproduction in sheep/edited by RB Land, DW Robinson.

Haresign W. (1985) The physiological basis for variation in ovulation rate and litter size in sheep: A review. Livestock Production Science 13, 3-20.

Xu S.S., Gao L., Xie X.L., Ren Y.L., Shen Z.Q., Wang F., Shen M., Eyϸórsdóttir E., Hallsson J.H., Kiseleva T., Kantanen J. & Li M.H. (2018) Genome-wide association analyses highlight the potential for different genetic mechanisms for litter size among sheep breeds. Frontiers in Genetics 9, 118.

Zhou Y., Zhou B., Pache L., Chang M., Khodabakhshi A.H., Tanaseichuk O., Benner C. & Chanda S.K. (2019) Metascape provides a biologist-oriented resource for the analysis of systems-level datasets. Nature Communications 10, 1523.

Round 2

Reviewer 1 Report

Dear authors,

I commend you for your scrupulous work done to explain what you did. I'm really satisfied by your answers. I appreciated that you followed my suggestions.

I have just one other minor comment:

lines 32-33: "One possible explain..." should be "One possible explanation..."

Best wishes

Author Response

Responds to reviewer 1

Dear authors,

I commend you for your scrupulous work done to explain what you did. I'm really satisfied by your answers. I appreciated that you followed my suggestions.

I have just one other minor comment:

lines 32-33: "One possible explain..." should be "One possible explanation..."

Best wishes

Reply: Thank you very much! We correct it.

Reviewer 2 Report

First of all I want to thanks the authors for their efforts in improving the manuscript.

In my opinion some of the explanations furnished in the revision note have to be included in the manuscript with advantages for the readers. More in details:

-about the studied breeds the authors wrote "these breeds excluding Texel are potential genetic resources to improve fecundity of other populations". This is an important information that should be included in the manuscript.

-about the software used for the graphical PCA representation: "The R package ggplot2 (https://ggplot2-book.org/index.html) was used for PCA visualization". This info should be included in the material and methods section.

-after the Bonferroni correction the p-value is usually named as q-value. This info could be specified in the manuscript. For CWAS I mean chromosome wide association.

Author Response

Responds to review 2

First of all I want to thanks the authors for their efforts in improving the manuscript.

In my opinion some of the explanations furnished in the revision note have to be included in the manuscript with advantages for the readers. More in details:

Reply: Thank you very much! We revised our manuscript according to your suggestions. Please see the revision and responds as followings.

-about the studied breeds the authors wrote "these breeds excluding Texel are potential genetic resources to improve fecundity of other populations". This is an important information that should be included in the manuscript.

Reply: Thank you. We put it on lines 66-67.

-about the software used for the graphical PCA representation: "The R package ggplot2 (https://ggplot2-book.org/index.html) was used for PCA visualization". This info should be included in the material and methods section.

Reply: Thank you. We add it to the end of section 2.1 in revised manuscript.

-after the Bonferroni correction the p-value is usually named as q-value. This info could be specified in the manuscript. For CWAS I mean chromosome wide association.

Reply: Thank you very much! We now describe it clearly. In this study we only focus on the outliers at whole-genome level to control the false positive rate.